# Chemical Control of *Corythucha arcuata* (Say, 1832), an Invasive Alien Species, in Oak Forests

**Flavius Bălăcenoiu** [1,2,*] **, Constantin Nețoiu** [2] **, Romică Tomescu** [1] **, Dieter Carol Simon** [1] **, Andrei Buzatu** [1,2] **, Dragoș Toma** [1,2] **and Ion Cătălin Petrițan** [1]

1    Faculty of Silviculture and Forest Engineering, Transilvania University of Brașov, Sirul Beethoven 1, 500123 Brașov, Romania; ro_tomescu2003@yahoo.fr (R.T.); disim@unitbv.ro (D.C.S.); andrei.buzatu@yahoo.com (A.B.); dragost93@gmail.com (D.T.); petritan@unitbv.ro (I.C.P.)
2    National Institute for Research and Development in Forestry "Marin Drăcea", Eroilor 128, 077190 Voluntari, Romania; c_netoiu@yahoo.com
*    Correspondence: flavius.balacenoiu@icas.ro

**Abstract:** In addition to the classic biotic and abiotic factors that have disrupted the health of forests throughout history, lately, the balance of forest ecosystems has been disturbed by different phenomena such as climate change, pollution, and, especially, biological invasions of invasive alien species. One of the alien species that has invaded Europe relatively quickly is an insect species of North American origin, the oak lace bug (*Corythucha arcuata* Say, 1832 Heteroptera: Tingidae). In the context of the rapid spread of infestations and the severity of attacks on oak trees in infested forests, this paper aims to assess measures to manage this species in the future. Namely, the effect of aerial chemical treatments on oak lace bug has been investigated with two influencing factors: the mode of insecticide action (contact and systemic) and the treatment volume (low volume and ultra-low volume). The experiment was conducted in two forests over a total area of 350 hectares. The results show that the reduction of the nymph population varied from 91% to 96%. However, the residual population was sufficient to allow differentiated re-infestations over time, more quickly after contact insecticide sprays (22 days after treatment) and slowly after systemic insecticide sprays (more than a month after treatment). This re-infestation time difference had implications on attack intensity as well, with stronger leaf discoloration observed in areas treated with a contact insecticide compared with those treated with a systemic insecticide.

**Keywords:** forest health; forest pests; chemical control; invasive alien species; oak lace bug

## 1. Introduction

The capacity of forests to accomplish multiple functions can be reduced by the harmful action of biotic and abiotic factors. Forests urgently need protection against the harmful action of these factors, especially in the light of the risks posed by global climate change, which is manifesting increasingly intensely in the last decade. People rely on healthy forest ecosystems for energy, building materials, and food, as well as services such as carbon storage, biodiversity management, and climate regulation [1]. During the lifespan of a forest, its health can be disrupted by biotic or abiotic factors, especially insect outbreaks, phytopathogens, wind, rainfall, or fire. Lately, in addition to these destabilizing factors, the health of forest ecosystems is endangered by different phenomena such as climate change and pollution, and, especially, biological invasions of invasive alien species (IAS) [1].

IAS are defined as organisms that have been introduced by humans, deliberately or accidentally, out of their natural range, that have multiplied and begun to exert negative effects on the new ecosystem [2–4]. The phenomenon by which they establish population, spread, and negatively affect native species is called biological invasion [2,3,5–7].

Against the background of increasingly intensive globalization, biological invasions have caused multiple negative effects on the economy and health, but especially on the

environment [7–16], becoming one of the factors implicated in the loss of biodiversity and changes in ecosystem services [4].

One of the alien species that invaded Europe relatively quickly is an insect species of North American origin, the oak lace bug (*Corythucha arcuata* Say, 1832 Heteroptera: Tingidae), which was first reported in Europe in 2000 in Northern Italy [17]. Two years later, in 2002, it was also reported in Switzerland [18] and Turkey [19]. Meanwhile, oak lace bug (OLB) expanded its range, reaching Eastern Europe. In 2012, it was first observed in Bulgaria [20], and in 2013, in Hungary [21], Croatia [22], and Serbia [23–25]. Since then, with the presence of the appropriate climatic conditions in Europe as well as a lack of natural enemies, OLB has experienced explosive growth. Thus, at present, the species has been reported in Russia [26], Romania [27,28], Albania [29], Slovenia [30], Bosnia and Herzegovina [31,32], France [33], Ukraine [29], Greece [29], Slovakia [34], and Austria [35].

After the first detection of the insect in many of the countries mentioned above, such as Turkey, Hungary, Bulgaria and Romania, the further invasion and expansion of the distribution of OLB was reported on the territory of the given country [36–39].

Both nymphs and adults of this species feed on the undersides of the leaves of host trees, usually oaks, by piercing the epidermis and drawing out the cellular sap material. The typical characteristics of OLB feeding are the spots resulting from small, separate stings (1–3 mm), which then grow and merge into gray-yellow spots on the upper surface of the leaf, resulting in chlorotic discoloration of the leaf [36,40,41].

In its area of origin, the OLB is not considered a significant pest, especially because there are a noteworthy number of native predators, such as *Hyaliodes vitripennis* (Say, 1832), *Deraeocoris nebulosus* (Uhler, 1832), *Erythmelus klopomor* (Triapitsyn), *Orius insidiosus* (Say, 1832), and syrphid larvae, which are used in integrated pest management [42–44]. In Europe, the way OLB has invaded the continent, as well as the extremely high population of OLB in consecutive years, suggests a high potential risk both ecologically and economically [41].

Given the wide range of varied host plants available to OLB, which feed on almost all Eurasian oaks [29], at least 30 million hectares of oak forest can be a suitable habitat in Europe [41].

Considering that thus far in Europe, no natural enemies have been identified that would significantly limit the population of OLB and thus limit the invasion of the species [41], it is necessary to study possible removal methods to prescribe experimentally tested control measures. In North America, where OLB is endemic, some insecticides are recommended as being effective [45–47]; however, no research has yet been performed using these. In Europe, small-scale experiments were carried out in a Serbian study [48] in which an 8 years old oak culture infested with OLB was treated with four chemical insecticides based on the active substances bifenthrin, buprofezin, thiamethoxam, and abamectin. The results showed that bifenthrin and thiamethoxam insecticides, which are neonicotinoids and pyrethroids, respectively, are highly efficient for OLB control. Based on these results, we set out to answer questions such as (i) whether the method chemical control used at the nursery level could be replicated at the forest level and (ii) of the two recommended insecticides, which is more effective, the neonicotinoid or the pyrethroid?

This research aims to assess the extent to which chemical control can limit the population of *Corythucha arcuata* and therefore limit the invasion of the species in Europe.

## 2. Materials and Methods

### 2.1. Study Site and Data Collection

The experiment was carried out in two forests (Forest A and B) in Romania, Giurgiu County (Figure 1). The forests are managed by the Bolintin Forest District and consist of Turkey oak (*Quercus cerris* L.), Hungarian oak (*Quercus frainetto* Ten.), and pedunculate oak (*Quercus robur* L.) stands or a mixture of these species, with small numbers of secondary species such as Norway maple, field maple, and European ash. The characteristics of the

two forests were extracted from the forest management plans of each and are summarized in Table 1.

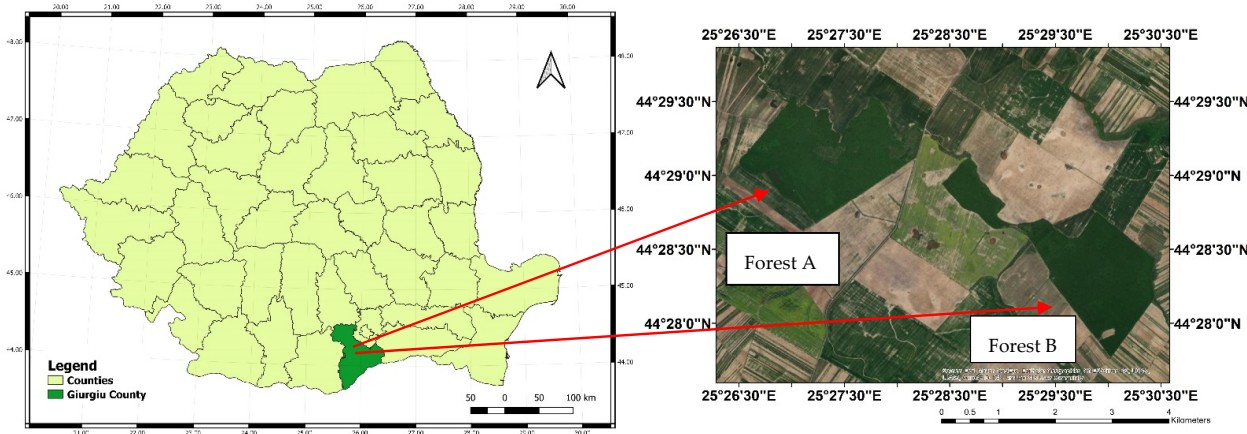

**Figure 1.** Location of the study area (Esri, Maxar, GeoEye, Earthstar Geographics, CNES/Airbus DS, USDA, USGS, Aero Grid, IGN, and the GIS User Community).

**Table 1.** Description of the two experimental forests.

| Characteristics | Forest A | Forest B |
|---|---|---|
| Coordinates | 44°29′06″ N 25°27′16″ E | 44°28′24″ N 25°29′53″ E |
| Area (ha) | 190 | 160 |
| Dominant forest species | *Quercus cerris, Q. frainetto, Q. robur* | *Quercus cerris, Q. frainetto, Q. robur* |
| Secondary forest species | *Acer platanoides, A. campestre* | *Acer platanoides, A. campestre* |
| Regeneration mode | natural regeneration (sprout) | natural regeneration (sprout) |
| Age (years) | 60 | 60 |
| Average diameter (cm) | 23 | 25 |
| Average height (m) | 19 | 20 |
| Crown density of the stand (%) | 70 | 90 |
| Standing volume (m$^3$ ha$^{-1}$) | 150 | 170 |

After monitoring OLB biology in previous years (data not yet published), we noticed adult activity starting in mid-April, egg-laying after mid-May, and the appearance of nymphs at the beginning of June. The experimental treatments were timed for the nymph stage. Nymphs lack wings; therefore, this was determined the most suitable time, given their inability to fly.

During the experiment, two factors were taken into account: the mode of action of the insecticide (contact or systemic) and the treatment volume (low-volume (LV) at 30 L/ha and ultra-low volume (ULV) at 3 L/ha).

To avoid factors overlapping and influencing each other, the sprays were applied in the following ways:

- Forest A was sprayed with a contact insecticide (Alfametrin 10CE) over an area of 190 ha. Both LV (0.1 L/ha commercial product in 29.9 L/ha water) and ULV (0.1 L/ha commercial product in 2.9 L/ha water) treatments were applied. Alfametrin 10CE is a synthetic pyrethroid insecticide that acts on harmful insects by contact and ingestion, and it is based on the active substance alpha-cypermethrin.
- Forest B was sprayed with a systemic insecticide (APIS 200 SE) over an area of 160 ha. Both LV (0.2 L/ha commercial product in 29.8 L/ha water) and ULV (0.2 L/ha commercial product in 2.8 L/ha water) treatments were applied. APIS 200 EC is a neonicotinoid insecticide that acts on harmful insects by systemic action, penetrating rapidly into the plants' leaves and acting on all stages of insect development, and it is based on the active substance acetamiprid.

The sprays were applied on 24 June 2020, when the majority of the OLB population was in the nymph stage of the first generation, but adults from the hibernating generation were also present. The medium air temperature at the time of the experiment was 25–26 °C. The manufacturer of Alfametrin recommends applying treatment when temperatures do not exceed 23–25 °C; therefore, the temperature should have no influence on the treatment efficacity. It is also worth noting that on the treatment day and the following days no precipitation was registered in the atmosphere.

The sprays were applied using a Kamov KA-26 helicopter (Figure 2) equipped with an Aera 660 GPS device (Garmin) for loading and tracking the preset route. Orientation during the flights was made based on the flight polygons loaded in the GPS equipment, with the distance between flight lines established at 35 m. Analysis of the flight routes recorded by GPS device established that the distance between flight lines ensured adequate coverage with the solution in both Forest A (Figure 2a) and Forest B (Figure 2b). The flight speed was between 100 and 120 km/h, and the altitude above the canopy was between 10 and 15 m. To spray the LV treatment, the helicopter was fitted with a PIRNA installation, which pumped the solution through nozzles mounted under the helicopter planes and evacuated it through slits in the nozzle body, creating a spectrum of spray droplets with diameters between 150 and 400 μ. To spray the ULV treatment, the helicopter was fitted with a Micronair AU5000 installation, which used a rotating woven wire gauze to produce approximately equal spray droplets with diameters of about 100 μ.

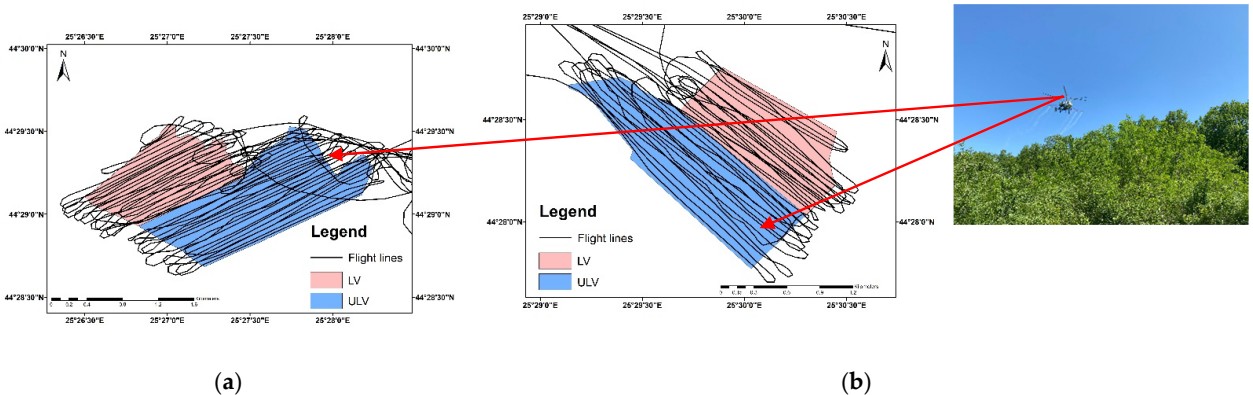

(**a**)                                         (**b**)

**Figure 2.** Analysis of the flight routes recorded by the GPS device in (**a**) Forest A (contact insecticide) and (**b**) Forest B (systemic insecticide).

To establish the effectiveness of each treatment, the density of nymphs on the day of application (Moment 0) was compared with the density after approximately 15 days (Moment 1). Next, to track the re-infestations, the density of nymphs was registered every 5–9 days until the beginning of August, when the nymphs of the second generation appear (Moments 2–5). We have a total of 6 moments during which monitoring was performed, as follows:

- Moment 0 (24 June 2020): the day of spraying
- Moment 1 (9 July 2020): checking the effectiveness of control methods
- Moment 2 (16 July 2020): checking re-infestations
- Moment 3 (24 July 2020): checking re-infestations
- Moment 4 (29 July 2020): checking re-infestations
- Moment 5 (5 August 2020): checking re-infestations

The nymphs population density was determined by collecting 30 leaves from 10 test trees for a total of 300 leaves at each monitoring moment. The number of OLB nymphs were counted, thus allowing the average number of nymphs on a leaf to be calculated. To ensure as uniform a distribution of leaves as possible, half of the leaves was collected from the lower canopy and the other half from the upper canopy of each tree.

To ensure a uniform working method, the test trees initially chosen were numbered so that the same 10 trees were used in subsequent checks.

Furthermore, because OLB nymphs and adults attack host trees by piercing the epidermis and drawing out the cellular sap material, we considered that another way to verify the effectiveness of control methods might be to measure the level of damage caused to insects in the active stage throughout the growing season. Thus, to assess the level of damage caused by *Corythucha arcuata* (Say, 1832), in the two forests, a network of observation points was evenly spaced over the area (Figure 3), in September, toward the end of the vegetation season, using a Juno SB GPS device (Trimble) with Terrasync software (Trimble), with an accuracy of five meters. Around each observation point, assessments were made on the attack intensity of the first 10 closest point trees (Figure 3). The method established by Tomescu et al. [39] was used to assess the intensity of the insect attack by visually evaluating the degree of foliage fading as a deviation from the normal color of the host oak leaves. Therefore, the intensity of the attack was evaluated by percentages of discoloration divisible by 5, ranging from 0% (not attacked) to 100% (strongly attacked). For clarity and homogeneity of the evaluations, we created a scale of discoloration (Figure 4).

The collected data regarding the attack intensity was statistically (see sub-chapter 2.2) and graphically (image analysis) interpreted on thematic maps based on the outline of the two forests. The thematic maps rendering was obtained through the ArcMap 10.5 software (ESRI) using ArcToolbox-Special Analyst Tools–Interpolation–IDW, assuming that each measured point had a local influence that diminished with distance and the measured values closest to the prediction location had more influence on the predicted value than those farther away. Subsequently, the 'renderers' raster data' function was applied for mapping, through the 'classified' method. To highlight the spatial differences between the observation points' attack by the degree of damage, we placed breaks at predefined threshold values for the percentage of discoloration taken as an average of the 10 trees in the point, as follows:

- 0–10%—very weak discoloration indicated by yellow ( )
- 11–25%—weak discoloration, indicated by pink ( )
- 26–50%—medium discoloration, indicated by green ( )
- 51–75%—strong discoloration, indicated by light blue ( )
- 76–100%—very strong discoloration, indicated by dark blue ( )

### 2.2. Data Analysis

To assess the effects of both applied insecticides (contact vs. systemic) and both treatment volumes (LV and ULV), and both treatment volume within each type of insecticide on the nymph population, a nonparametric test was performed. Additionally, to assess the effects of each type of insecticide on the lower canopy and upper canopy, and to compare the effects of both types of insecticide within each canopy layer on the nymph population, the same nonparametric test was applied. Before testing, Shapiro–Wilk and Levene's tests were used to check the normality of data and homoscedasticity of variances, respectively. Because no data complied with the requirements of parametric tests such as the two-way ANOVA (analysis of variance), we applied the Mann–Whitney U test. All statistical significance tests were performed using STATISTICA 12.1 (StatSoft, Inc., Tulsa, OK, USA).

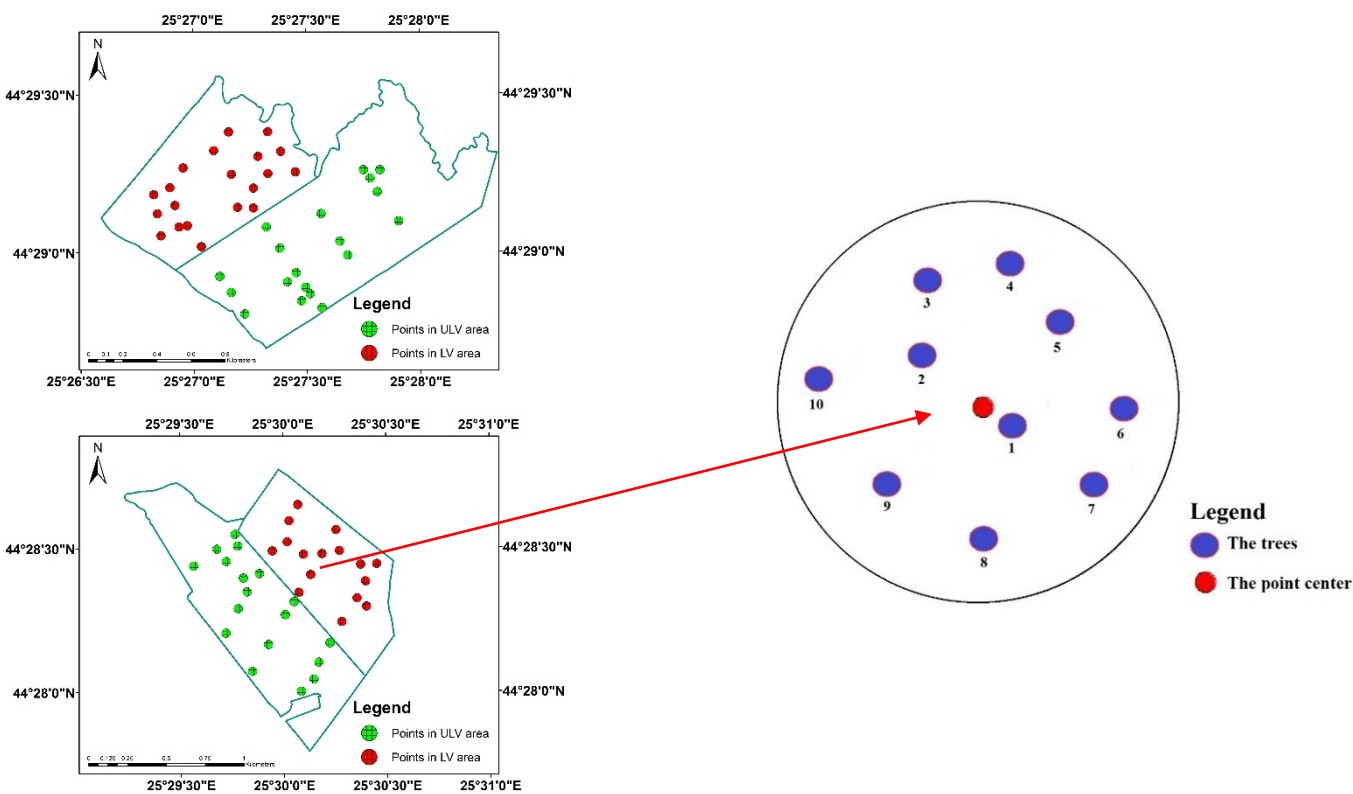

**Figure 3.** The network of observation points established to assess the level of damage caused by *Corythucha arcuata*.

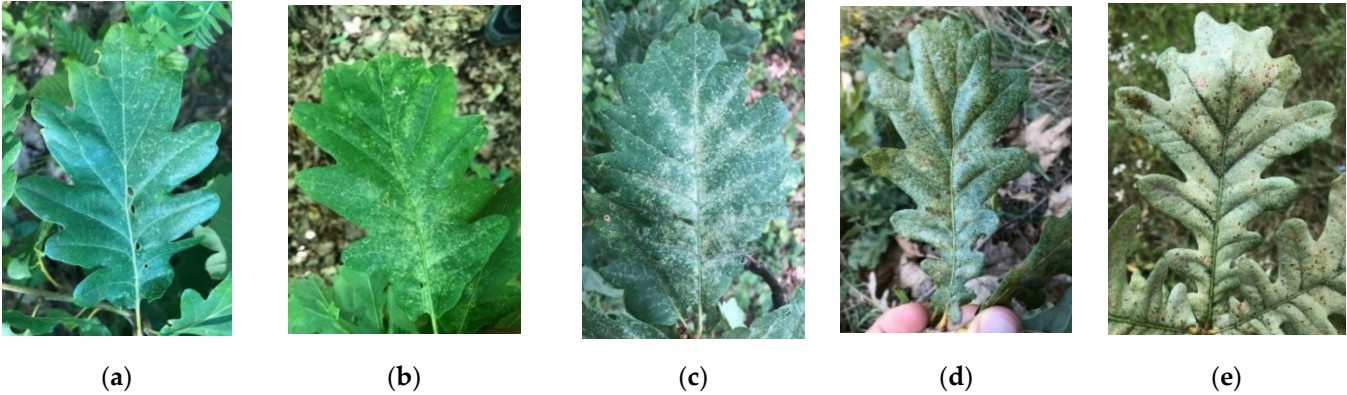

**Figure 4.** Scale showing different attack intensities of *Corythucha arcuata* nymphs and adults on oak leaves: (**a**) 5%; (**b**) 25%; (**c**) 50%; (**d**) 75%; and (**e**) 100%.

## 3. Results

### 3.1. Effects of Applied Insecticide and Treatment on Nymphs' Density

At the population level (Figure 5a,b), a high density of nymphs was observed at Moment 0, before treatment. The nymph population immediately after spraying was reduced by up to 95% in both forests under both LV and ULV treatments.

Next, there was a slight re-infestation over time, starting at the end of July and being much more apparent in Forest A (Figure 5a), which was sprayed with a contact insecticide, than in Forest B (Figure 5b), which was sprayed with a systemic insecticide. In both forests, the population remained at a certain level and never rose to the initial level.

Given that at Moment 0 (24 June 2020), the average number of nymphs on a leaf was calculated between 4.26 and 6.51, and no significant differences in population density were observed among the four experimental areas in the two forests (Figure 6a), we presumed

that all four areas in the experiment were homogeneously infested before the experiment. In checking the treatment effectiveness, at Moment 1 (09 July 2020), we found that the nymph population density was reduced to 0.24–0.50 nymphs per leaf, without significant differences among the four experimental areas (Figure 6b). Thus, the population was reduced by 91–96%.

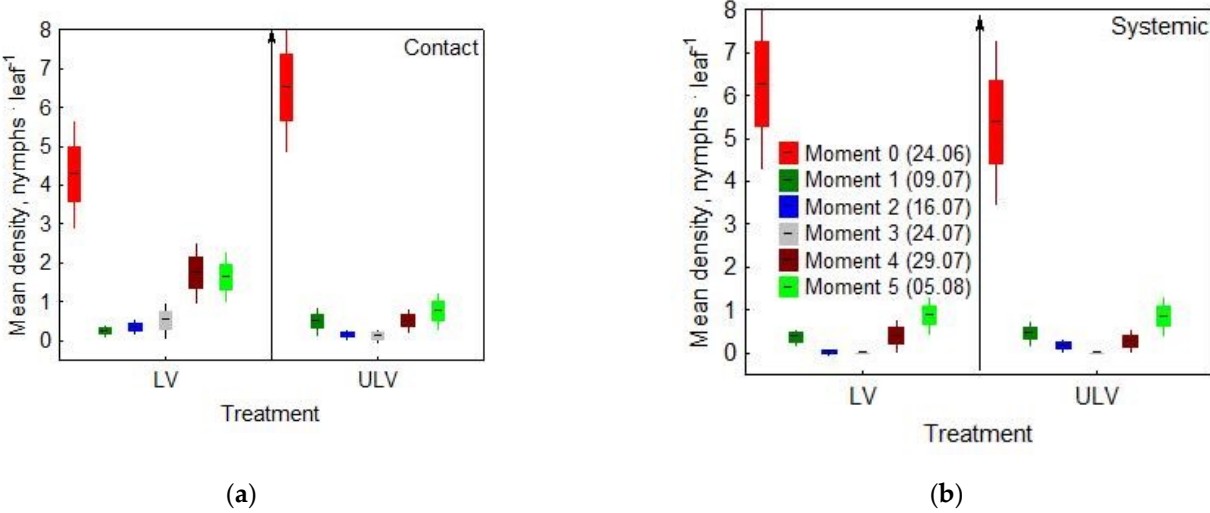

**Figure 5.** Dynamics of nymphs over the observed period (Moment 0, before spraying, and M1–M5, five moments after spraying) separated by treatment volume (LV or ULV) and insecticide type: (**a**) contact insecticide; and (**b**) systemic insecticide.

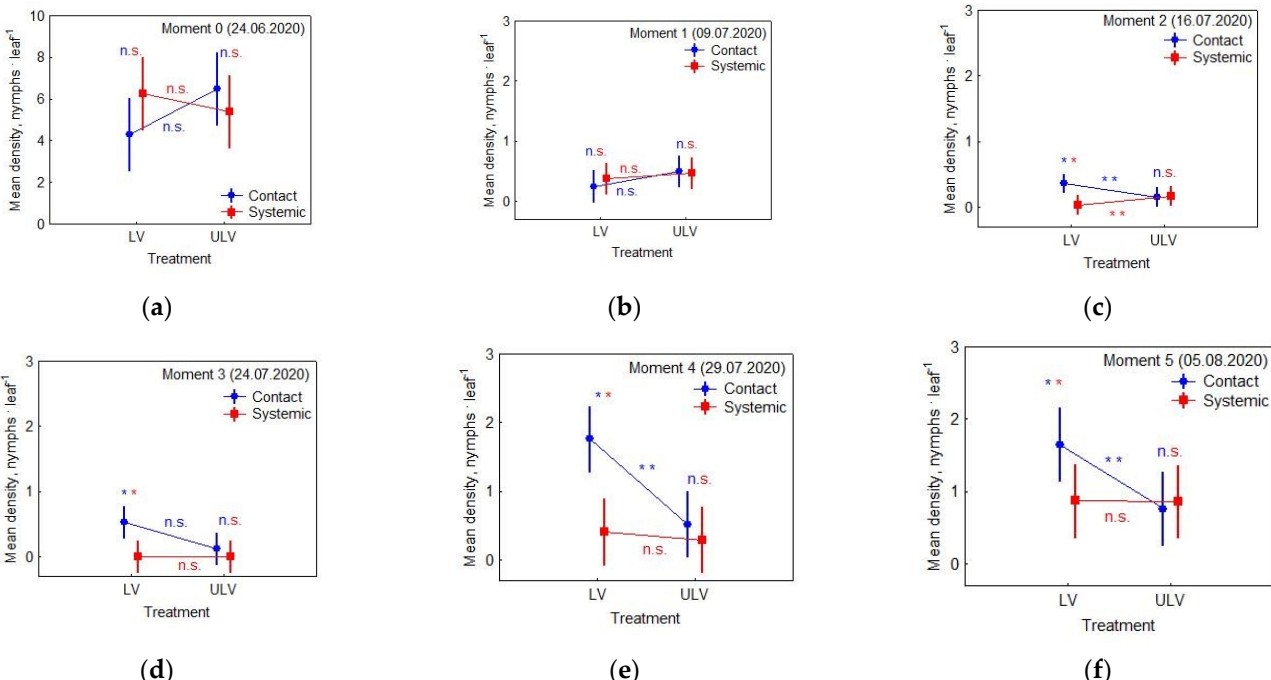

**Figure 6.** Effects of applied insecticide type (contact vs. systemic) and treatment volume (LV vs. ULV) on nymph density at each measurement time: (**a**) M0 (before spraying)-24 June 2020; (**b**) M1-09 July 2020; (**c**) M2-16 July 2020; (**d**) M3-24 July 2020; (**e**) M4-29 July 2020; and (**f**) M5-05 August 2020. ** = significant differences (U test, $p < 0.05$); n.s. = nonsignificant.

We began to record the re-infestation of the experimental area starting with Moment 2. At Moment 2 (Figure 6c), there was a slight increase in the density of nymphs in the Forest A–LV area, while the other three areas (Forest A-ULV, Forest B-LV and Forest B-ULV) showed a clear decrease in nymph population.

At Moment 3 (Figure 6d), while the Forest A-LV population noticeably increased, the other three areas showed decreases in nymph density to values close to 0 nymphs per leaf. At this time, there were significant differences between the population density in contact-LV areas and systemic-LV areas.

At Moments 4 and 5 (Figure 6e,f) the hypothesis made in previous moments regarding the strong re-infestation of the Forest A-LV area was confirmed, and an attempted re-infestation was also observed in Forest A-ULV but was not statistically assured. Furthermore, in Forest B–LV and Forest B-ULV, there were slight increases in the population density without statistical differences between these areas. At both moments, there were significant differences between the population density in Forest A-LV and the population density in Forest B-LV and Forest A-ULV.

To identify the main reason Forest A was re-infested, we compared the nymph population density at the upper and lower canopy after systemic and contact treatments.

The infestation was initially homogeneous at both upper and lower levels of the canopy in both forest B (Figure 7a). Furthermore, even after treatment, at Moment 1 (Figure 7b), nymph population density decreased consistently at both canopy levels in both forests.

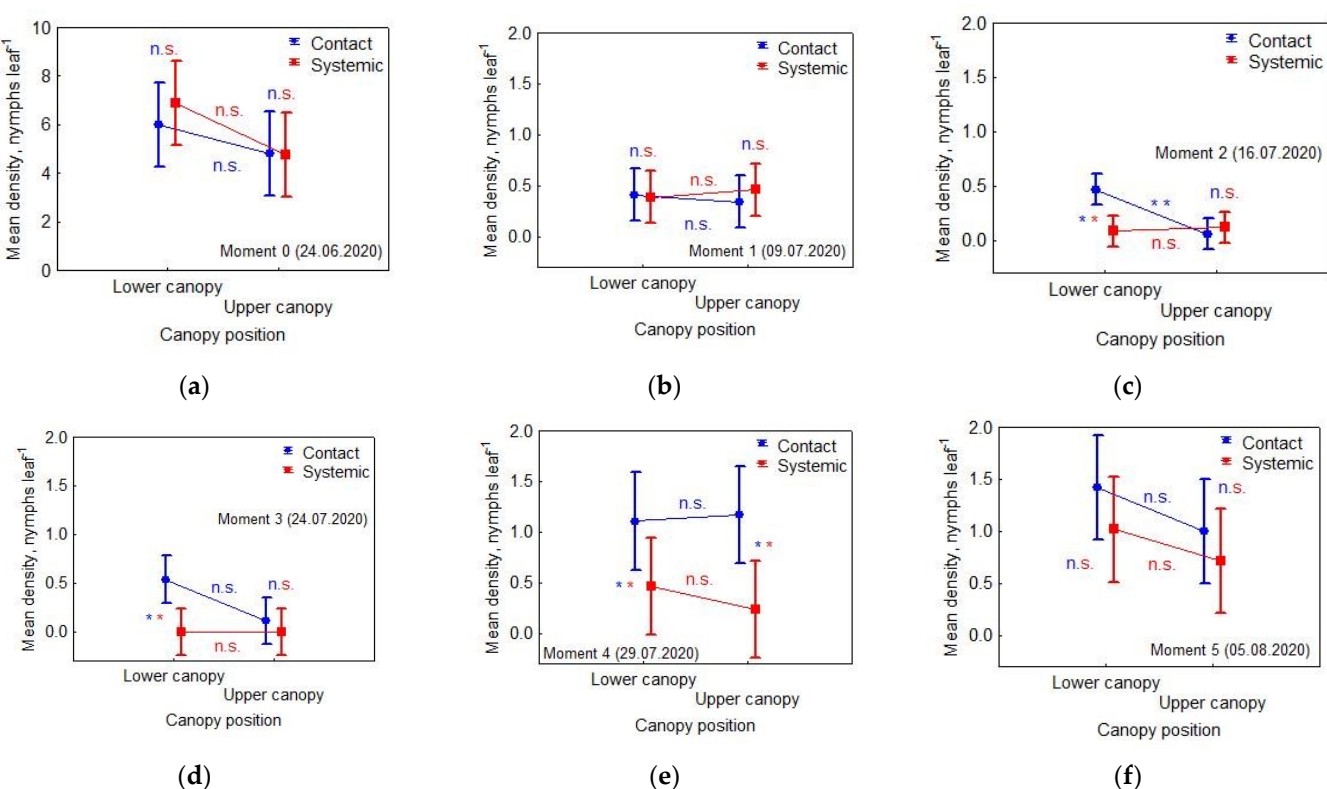

**Figure 7.** Effects of applied insecticide type (contact vs. systemic) on nymph density by leaf position (lower canopy vs. upper canopy) at each measurement time: (**a**) M0 (before spraying)-24 June 2020; (**b**) M1-09 July 2020; (**c**) M2-16 July 2020; (**d**) M3-24 July 2020; (**e**) M4-29 July 2020; and (**f**) M5-05 August 2020. ** = significant differences (U test, $p < 0.05$); n.s. = nonsignificant.

At Moment 2 (Figure 7c), there was a clear increase in the lower canopy population in Forest A. This increase was statistically significant both compared with the population density in the upper canopy in Forest A but also with that in the upper canopy in Forest B.

At Moment 3 (Figure 7d), the population density in Forest B insecticide remained low, with close to 0 nymphs per leaf and with no statistical differences between the two canopy levels. However, in Forest A, the population density increased even in the upper canopy, this time without any statistical differences between the two canopy areas. At the same

time, the population level in the lower canopy in Forest A was statistically significantly higher than that in Forest B.

At Moment 4 (Figure 7e), the re-infestation of Forest A was apparent through homogenized population density at both canopy levels. Similarly, there was a slight increase in population density with similar values between both canopy levels, in Forest B, although the population values were statistically significantly lower than those in Forest A.

At Moment 5 (Figure 7f), we continued to observe an increase in population density in both forests, without statistical differences between the indicators, but with lower population values for the area treated with a systemic insecticide.

### 3.2. Attack Intensity Assessment at the End of Growing Season

The attack intensity of OLB nymphs and adults at the forest level (Figure 8) shows that Forest A experienced the highest attack intensities, at 80.9% in the ULV area and 76.5% in the LV area (no significant differences between them). Forest B showed statistically significant differences both depending on the application method and compared with Forest A, at 53.2% in the ULV area and 27.6% in the LV area.

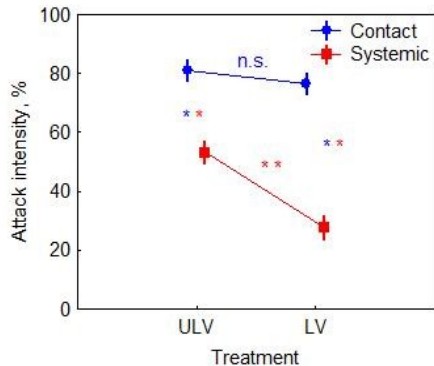

**Figure 8.** Attack intensity assessment at the end of the growing season (three months after treatment) by applied insecticide type and treatment volume. ** = significant differences (U test, *p* < 0.05); n.s. = nonsignificant.

The thematic maps show higher discoloration intensity in Forest A (Figure 9a) compared with Forest B (Figure 9b). Moreover, the treatment volumes resulted in higher discoloration intensity ULV areas compared with LV areas.

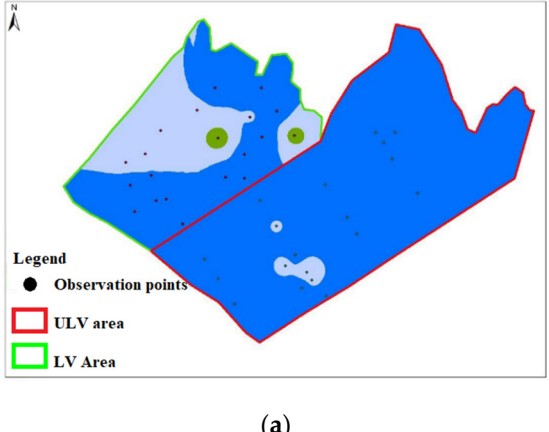

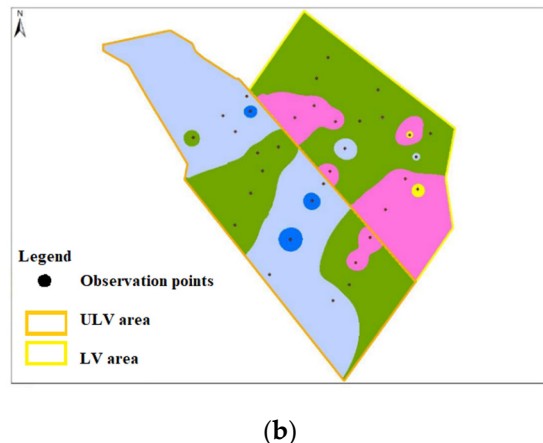

(**a**)　　　　　　　　　　　　　　　　　　　　　　　　　(**b**)

**Figure 9.** Thematic maps showing attack intensity by insecticide type and treatment volume: (**a**) contact insecticide (Forest A); and (**b**) systemic insecticide (Forest B). ▮ very weak discoloration (0–10%); ▮ weak discoloration (11–25%); ▮ medium discoloration (26–50%); ▮ strong discoloration (51–75%); and ▮ very strong discoloration (76–100%).

## 4. Discussion

Considering the initial nymph population, we can conclude that the experimental treatments were 91–96% effective at Moment 1. If we look at the short term, immediately after spraying, our results at the stand level match the results at the nursery level [48]. Both insecticide types and treatment volume were highly effective. However, in the long term and regarding the dynamics of the population as a whole, we observed a re-infestation tendency at the end of July (Moment 4, 29 July 2020) in all four experimental areas, overlapping with the development of a new generation of nymphs, as observed in OLB populations in Italy [49]. Consequently, this re-infestation phenomenon in all experimental areas can be explained by the particular phenology of OLB and the hypothesis that the residual population was sufficient to allow re-infestations over time.

Forest B showed an exponential increasing in population (the population density increased sharply at Moment 4), while Forest A showed a linear increase (the population density increased steadily from Moment 2). Therefore, Forest A registered population increases immediately after spraying, which leads us to conclude that other factors affected the experiment and, as a result, it was not successful.

One of the hypotheses to explain Forest A re-infestation could be the action mode of the insecticide. The contact insecticide used may have performed best in the upper part of the canopy, where the toxic particles reached. According to this hypothesis, the re-infestation happened in the lower canopy (Figure 7), where the toxic substance would not reach. This could be studied in future research by measuring the drop dispersion pattern in the upper and lower canopy.

Another possible justification for the increase in nymph population at Moment 2 (16 July 2020) in Forest A is the known fact that contact insecticides generally persist for a very short period. There are studies [50,51] in which experimental injections of cypermethrin-based contact insecticide into the trunks of affected trees have shown longer persistence against bark beetles. The injection method is even recommended for systemic insecticides used for lace bug control [45]. Although this option is considered highly effective in the native habitats of lace bugs, we believe this procedure would be difficult to implement, given the large size of infested oak areas in Europe.

The highest nymph mortality was recorded in Forest B-LV, which received better dispersion of particles and a slow-acting insecticide. In Forest A, although mortality was high immediately after the application, the effect diminished over time and allowed for faster re-infestations and higher population densities compared with Forest B.

The analysis of the attack intensity of OLB nymphs and adults shows that both the action mode of the insecticide used (contact or systemic) and the treatment volume (ULV or LV) have a significant impact on the degree of foliage discoloration.

The registered discolorations toward the end of the growing season were more reduced after a LV application of a systemic insecticide, which offered a better spread of toxic particles inside the canopy and thus ensured better foliage protection.

The significant difference in damage recorded in the two forests after treatment can be explained by the different times of re-infestation. Forest A was re-infested at Moment 2 and Forest B only at Moment 5. During this entire time, in Forest A, both adults and nymphs OLB fed continuously.

In addition to the damage caused to the foliage, if we consider that 80% of the radial growth of oaks occurs in the first part of the growing season, until the end of July [52–54], the treatments used on Forest B offer another advantage in protecting the trees during the period of greatest growth, allowing re-infestation to begin in early August.

Therefore, although all four treatments seemed effective immediately after spraying, by the end of the experiment, we could conclude that the systemic insecticide, especially applied at LV, proved more effective than the contact insecticide, judging by the OLB damage caused by the end of the growing season. Our results were therefore in agreement with the hypothesis Paulin et al. [41] put forward, that systemic insecticides are likely to be

the best treatments options because both larvae and adults are relatively concentrated on the undersides of the leaves.

However, if we consider that in Europe, 27 oak species have been proven suitable hosts for OLB [29], it is difficult for the results of this experiment to be put into practice. For example, as of autumn of 2019, it is estimated that OLB have infested over 1.7 million hectares of forest in: Croatia, Hungary, Romania, Serbia, and Russia (in Europe) [41].

Furthermore, if we take into account the recommendations of Dreistadt and Perry [45] and Shetlar [47] that, for adequate control to prevent re-infestations, the undersides of leaves must be thoroughly wet and spraying with toxic substances should be repeated, we conclude that chemical control methods are not economically justifiable. However, considering that lately, citizens have complained of discomfort caused by OLB stings or bites [55], chemical control might still be applied in isolated cases, such as frequented parks, park forests, private forests, forests of tourist interest, gardens, or isolated trees. In these cases, based on the results of this study, we recommend the use of a systemic insecticide, applied with discernment due to its persistence over time that could affect natural OLB enemies or pollinators.

It is worth noting that in North America, OLB populations are not significant risks to oak forests due to different control tactics such as cultural methods and, especially, biological methods. Only after these methods are chemical methods considered [45–47]. Based on this model of good practice, we support the statement of Paulin et al. [41] that the only viable OLB control method in the large forests recently invaded in Europe is classical biological control, but this must also be preceded by isolated experimental studies on ecological balance. Moreover, this method would be consistent with the general public's preference for biological control of invasive species over chemical control, as reported by Japelj et al. [56].

## 5. Conclusions

In the short term, the population density of OLB nymphs was reduced in all four experimental areas by 91–96%.

However, in Forest A, sprayed with a contact insecticide, re-infestation was observed starting 16 July 2020, 22 days after treatment. In Forest B, sprayed with a systemic insecticide, the re-infestation was observed more than a month later, starting 05 August 2020.

The analysis of attack intensity shows stronger discoloration in Forest A (sprayed with contact insecticide) compared with Forest B (sprayed with systemic insecticides). Furthermore, when using the same type of insecticide, stronger discoloration was observed in ULV application areas compared with LV application areas.

The only viable option to control OLB in large forests is classical biological control, but this must be preceded by isolated experimental studies on ecological balance.

**Author Contributions:** Conceptualization, F.B., C.N. and R.T.; methodology, F.B. and C.N.; software, F.B., A.B. and I.C.P.; validation F.B., C.N., R.T., A.B., D.T., D.C.S. and I.C.P.; formal analysis F.B., A.B. and I.C.P.; investigation, F.B., A.B. and D.T.; resources, F.B., C.N. and A.B.; data curation, F.B., D.T. and A.B.; writing—original draft preparation, F.B.; writing—review and editing, C.N., R.T., A.B., D.C.S. and I.C.P.; visualization, R.T. and D.C.S.; supervision, C.N.; project administration, F.B.; funding acquisition, R.T. All authors have read and agreed to the published version of the manuscript.

**Funding:** The data collection for this paper was carried out within the project PN 19070201 "Assessment of the risk of new species of harmful insects with potential for outbreak of deciduous forests in Romania", and the writing was made in the project PN 19070202 "Improving methods of surveillance and control of harmful insects using modern technologies".

**Acknowledgments:** We thank the staff of the Forest Protection Department–ROMSILVA, Giurgiu Forest District and INCDS 'Marin Drăcea' for their assistance in fieldwork. Likewise, we acknowledge that this study is part of Flavius Bălăcenoiu PhD thesis (Bioecology of the invasive alien species *Corythucha arcuata* (Say.) in Romania). We would also like to thank the two anonymous reviewers who made suggestions for improving the work.

**Conflicts of Interest:** The authors declare no conflict of interest. The funders had no role in the design of the study; in the collection, analyses, or interpretation of data; in the writing of the manuscript, or in the decision to publish the results.

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
