# Peer review of "Chemical Control of Corythucha arcuata (Say, 1832), an Invasive Alien Species, in Oak Forests"

_forests, doi:10.3390/f12060770_

Round 1

Reviewer 1 Report

The issue, namely experimenting different options of chemical control on invasive insects, is of major significance for European forests, increasingly affected by alien parasites.The paper is clearly written and relatively easy to read. The current form of the paper has not substantial problems.

The graphs of figures 4a and 4b should be improved, as it is not easy to appreciate the values in the y axis; it could be used a scale value of 1 or 0.5 instead of 2.

Finally, the Conclusions should be expanded with considerations about the biological control method that previously in the paper was defined as the preferred one by “the general public”; however, apparently no natural enemies are active in European forests at present.

Here are some more detailed comments and corrigenda:

Line 108, Table 1: the Density in the Characteristics column is not clear how it is defined (individuals per ha refer to trees?); please clarify

Line 173: Model of GPS and accuracy

Line 186: If the authors used ArcMap by ESRI they should cite the software and company

Line 186-189: The authors should better describe the tools used and why they used them.

Line 208: The term “removal” is not appropriate in my opinion, better “treatment” or “spraying”

Line 304: increasing?

Line342: “radial growth of oaks is part of the first part……”, please clarify and correct.

Line 364: “with carefully….”please clarify and correct.

Line 382: In my opinion the term “experiment” in this sentence is not used correctly; treatment or spraying would be more appropriate.

Reviewer 2 Report

The submitted manuscript dealt with chemical control of Corythucha arcuata. The experiments were performed well, and the manuscript described with scientific soundness. Therefore, I recommend it to be published. Before publication, the following needs to be addressed.

Aerial insecticide application in forest are different from that in agriculture. Therefore, It would be recommend to describe more details on insecticide application such as spray nozzle, flight speed, aerial height, drop dispersion pattern and so on.

For the agricultural crops, the amount of 3 L/ha could be covered all the crops depending on the kind of crop.

However, it is hard to agree that 3L/ha in oak tree forest could covered all the trees. If the authors checked the droplet, describe the drop dispersion pattern (upper and lower canopy).

In ULV treatment, the insecticide was 3 folds diluted. Isn’t there any toxicity to tree? Describe it.

Line 310: The authors said “In this way, the re-infestation happened on the lower canopy where the toxic substance did not reach (Figure 6).” It is hard to agreed. If the insecticide did not reach the lower canopy, density of nymphs would be expected not to be changed after application (Fig 6b). It would be solved by the check droplet.

Contact and systemic insecticide: systemic insecticide also works by contact to insect; therefore, I recommend to change contact insecticide as non-systemic insecticide.

Line 133-134: the experiment was performed at the temperature between 24⁰C and 29⁰C. However, according to manufacture’s instruction, alfametrin is advised to perform the treatments when the temperatures do not exceed 23-25 °C. Isn’t there any effect of the temperature in insecticidal effect of alfametrin? Describe about it.

Line 293: Describe the time point of the effectiveness. Does it obtained from the result of Moment 1?

Table 1. Density (individuals ha-1) 0.7, 0.9 : Dose it mean 0.7 tree in ha? Please check the unit.

Figure 2, 3 and 8

ULV and LV area of Forest A in Figure 2 and Forest A in Figure 3 and 8 are opposite. Which one is correct?

Figure 3 Right: Was the point center (Red circle) is applied by LV and was periphery 10 points (light green) applied ULV? It is needed to refine description.

Minor points

Line 62: change to Ukraine

Line 123: alfa-Cypermethrin – change to alpha-cypermethrin

Line 130: Acetamiprid - acetamiprid
